

# Drought correlates with reduced infection complexity and possibly prevalence in a decades-long study of the lizard malaria parasite *Plasmodium mexicanum*

Allison Neal[1], Joshua Sassi[1] and Anne Vardo-Zalik[2]

[1] Norwich University, Northfield, VT, United States
[2] Pennsylvania State University York, York, PA, United States

## ABSTRACT

Microparasites often exist as a collection of genetic 'clones' within a single host (termed multi-clonal, or complex, infections). Malaria parasites are no exception, with complex infections playing key roles in parasite ecology. Even so, we know little about what factors govern the distribution and abundance of complex infections in natural settings. Utilizing a natural dataset that spans more than 20 years, we examined the effects of drought conditions on infection complexity and prevalence in the lizard malaria parasite *Plasmodium mexicanum* and its vertebrate host, the western fence lizard, *Sceloporus occidentalis*. We analyzed data for 14,011 lizards sampled from ten sites over 34 years with an average infection rate of 16.2%. Infection complexity was assessed for 546 infected lizards sampled during the most recent 20 years. Our data illustrate significant, negative effects of drought-like conditions on infection complexity, with infection complexity expected to increase by a factor of 2.27 from the lowest to highest rainfall years. The relationship between rainfall and parasite prevalence is somewhat more ambiguous; when prevalence is modeled over the full range in years, a 50% increase in prevalence is predicted between the lowest and highest rainfall years, but this trend is not apparent or is reversed when data are analyzed over a shorter timeframe. To our knowledge, this is the first reported evidence for drought affecting the abundance of multi-clonal infections in malaria parasites. It is not yet clear what mechanism might connect drought with infection complexity, but the correlation we observed suggests that additional research on how drought influences parasite features like infection complexity, transmission rates and within-host competition may be worthwhile.

# INTRODUCTION

Malaria parasites (*Plasmodium*, phylum Apicomplexa) reproduce asexually as genetically-distinct clones of haploid cells within the vertebrate host's blood cells. The majority of research on these parasites for both humans (*e.g.*, *Read & Taylor, 2001*; *Cui et al., 2003*; *Barry et al., 2013*; *Fola et al., 2017*; *Koepfli & Mueller, 2017*) and wildlife hosts

Corresponding author
Anne Vardo-Zalik, amv12@psu.edu

(*Vardo & Schall, 2007*; *Jarvi, Farias & Atkinson, 2008*; *Fong et al., 2014*; *Hicks & Schall, 2014*) reveals that many infections consist of multiple genetic clones (herein termed complex infections or multi-clone infections). Complex infections are presumed to be established primarily through transmission events, with either a single host receiving sequential vector bites or a single bite transferring multiple clones (termed collective transmission of clones; *Read & Taylor, 2001*; *Vardo-Zalik, 2009*). Additionally, albeit more rare, new clones may arise during the course of infection *via* mutation, especially when inadequate drug-treatment is used (*Cottrell et al., 2014*). Infection complexity can have important implications for the progression of pathology suffered by the host (disease progression) and lifetime virulence (*de Roode et al., 2005*; *Bell et al., 2006*; *Wargo et al., 2007*; *Vardo-Zalik & Schall, 2008*; *Vardo-Zalik & Schall, 2009*; *Vardo-Zalik, 2009*; *Kiwuwa et al., 2013*). For example, *Vardo-Zalik & Schall (2009)* found that diverse infections of *Plasmodium mexicanum* had more variable growth rates and parasitemias than their single-clone counterparts. Additionally, lizards infected with >3 genetic clones of the parasite had reduced virulence (measured by blood glucose and hemoglobin concentrations) compared to infections with fewer clones (*Vardo-Zalik & Schall, 2008*).

Thus, because complex infections in malaria parasites are common and can lead to medically and epidemiologically important outcomes, understanding what factors influence within-host diversity is an important goal for disease ecology. Transmission events are a major source of multi-clone infections, so infection complexity is often viewed as a correlate of transmission intensity (*e.g.*, *Tusting et al., 2014*). Any mechanism that alters overall transmission could therefore also affect the number of cohabitating clones within an infection. One such mechanism would be the onset of drought. Vector borne parasites such as *Plasmodium* depend on arthropod vectors for transmission, and many parasite vectors need water or humidity for reproduction and survival, including mosquitoes and sand flies (*Confalonieri, Margonari & Quintão, 2014*). As a result, limitations on water availability are likely to affect transmission dynamics, potentially leading to changes in prevalence and complexity of infection (COI). Climatologists predict that severe droughts are likely to increase in certain areas as a result of climate change (*Patz, Engelberg & Last, 2000*; *Patz et al., 2000*; *Cardenas et al., 2006*; *He, Russo & Anderson, 2017*; *Pokhrel et al., 2021*), meaning research on the impacts of drought will be crucial to planning for future disease outbreaks.

Thus far, much empirical research on the impacts of drought on parasite ecology has focused on disease prevalence/incidence as well as vector population size and distribution (see *Gagnon, Smoyer-Tomic & Bush, 2002*; *Brown, Medlock & Murray, 2014*), but to our knowledge, no study has reported on the effects of drought on infection complexity. While previous studies on human malaria have shown changes in COI during wet and dry seasons in Africa, these studies did not specifically measure for drought conditions (for example: *Gnagne et al., 2019*; *Collins et al., 2022*). Infection complexity, although often correlated with disease prevalence (*Wanji et al., 2012*; *Tusting et al., 2014*; *Gatei et al., 2015*; *Fola et al., 2017*), can be high despite overall low prevalence and *vice versa* (*e.g.*, *Barry et al., 2013*; *Gunawardena et al., 2014*; *Fola et al., 2017*; *Koepfli & Mueller, 2017*). Given that infection complexity plays a key role in pathogen biology, we set out to determine the

effects of drought on the genetic diversity of the saurian malaria parasite, *Plasmodium mexicanum*, at our field site in northern California, perhaps independent of changes in prevalence. Using genetic data spanning over two decades and prevalence and rainfall/drought data from four decades, we set out to answer the following questions for the malaria parasite *P. mexicanum*: (1) over a 20-year period, is drought a good predictor of malaria infection complexity in this parasite-host system? (2) are any changes in COI accompanied by corresponding changes in the overall prevalence of malaria infection, as would be predicted by standard transmission models?

## MATERIALS AND METHODS

### Collection of samples

We utilize data from a long-term study at the University of California's Hopland Research and Extension center (HREC), located in southern Mendocino County, California. Data on the infection status of lizards were available for the full range of dates sampled, 1978 to 2016, while samples for genetic analysis were only regularly collected starting in 1996. During the roughly four decades of this study, over 25,000 lizards have been collected from more than 100 sites at the field station. A subset of these sites (termed 'annual sampling' sites) where the parasite is relatively common has been consistently sampled to compare malaria prevalence and diversity over the full range of collection dates. In most years, a target of 50 lizards per site is collected from these annual sampling sites.

Lizards were collected by noosing and a small sample of blood taken through a toe clip; this blood was used to produce a thin smear that, when stained with Giemsa, can be used to identify infected lizards (below) and more recently, some blood has also been stored on filter paper for genetic analysis (below). In addition to taking a blood sample, the lizards' capture site, sex, and size (snout-to-vent length, SVL, a standard measure that correlates with age) were recorded. Over this timeframe, the vast majority of lizards collected were released at their point of capture within 24 h. Lizards were marked (*e.g.*, with white-out) prior to release to ensure that the same lizard was not accidentally re-sampled within a given year. Some years, each lizard was also given a unique toe-clip combination as a more permanent identifier; these can be used to identify lizards recaptured in a subsequent year. Recapture in subsequent years is rare because lizards are relatively short-lived (most are found fewer than three consecutive years, *Eisen, 2001*), so it is reasonable to treat every captured lizard as an independent data point for analysis.

In 2013–2014, California experienced its worst drought on record, though notable droughts also occurred in 1976–1977 (just before lizard malaria research started at HREC) and in the late 1980s to early 1990s (*Swain et al., 2014*). Examination of malaria prevalence before and after this historic drought as well as over the full range of years for which data are available should therefore provide insight into the effects of drought on the prevalence of a lizard malaria parasite.

### Animal use and field approvals

Because the data analyzed in this paper were drawn from many studies conducted by many researchers over many years, we were not able to locate individual IACUC or CA collection

permission numbers for all of the past studies, but we are confident that the collections were made in accordance with internationally accepted guidelines. All collections were made by students or former students of Jos. J. Schall in accordance with collecting permits from the state of California and IACUC protocols, most of which were granted by the University of Vermont. The methods in earlier IACUC protocols were virtually identical to those in the protocols supporting the most recent years of this research: IACUC protocols 45057 and 36698, granted by The Pennsylvania State University. Scientific Collecting Permits from the state of CA were also procured over the course of this study, with the more recent samples utilizing permit SC-7971. This research was performed at the Hopland Research and Extension Center, in Hopland, CA and the work associated with this project was approved by their research board (project 65-15).

## Identification of infections

Infections were identified based on a 3–6 min scan of Giemsa-stained thin blood smears. If any parasites were detected in the blood during this time, the lizard was recorded as infected. Only one malaria parasite (*P. mexicanum*) is known to infect the *S. occidentalis* at HREC. There are a few other blood parasites that were occasionally seen (*e.g.*, *Schellackia* sp., haemogregarines), but they are easily differentiated *via* microscopy. Past research comparing the number of infections identified *via* microscopic scans *vs* PCR amplification with species-specific primers revealed that most infections have a high enough density of parasites in the blood to be identified *via* microscopy (*Perkins, Osgood & Schall, 1998*).

## Estimation of COI

Since 1996, a few drops of blood have been regularly collected from each lizard and stored frozen on filter paper for genetic analysis. DNA was extracted from these samples and amplified for up to four microsatellite loci: Pmx306, Pmx732, Pmx747 and Pmx839 (*Schall & Vardo, 2007*). Microsatellites are repetitive regions of DNA that are thought to mutate primarily *via* replication slippage, generating alleles that differ in length. Mutation rates are relatively high, so individuals can often be differentiated by their microsatellite alleles. Malaria parasites replicate asexually in the blood of their vertebrate host, so infections may contain one or more clonal lines ('clones'). Conveniently, all blood stages are haploid, so the number of distinct clones can be estimated by examining the number of alleles at a single locus; single-clone infections will have only a single allele at any locus, whereas two-clone infections will have up to two alleles, and so on. The maximum number of alleles observed at any locus therefore represents the minimum number of clones present (*i.e.*, the COI) in a given infection (see *Vardo & Schall (2007)* for methods on allele scoring).

## Drought and rainfall data

The US Drought Monitor (USDM) is produced by the National Drought Mitigation Center at the University of Nebraska Lincoln in collaboration with the National Oceanic and Atmospheric Administration (NOAA) and the U.S. Department of Agriculture (USDA). It rates the severity of drought throughout the United States based on various indicators including soil moisture, streamflow and precipitation. Ratings are 0 (no

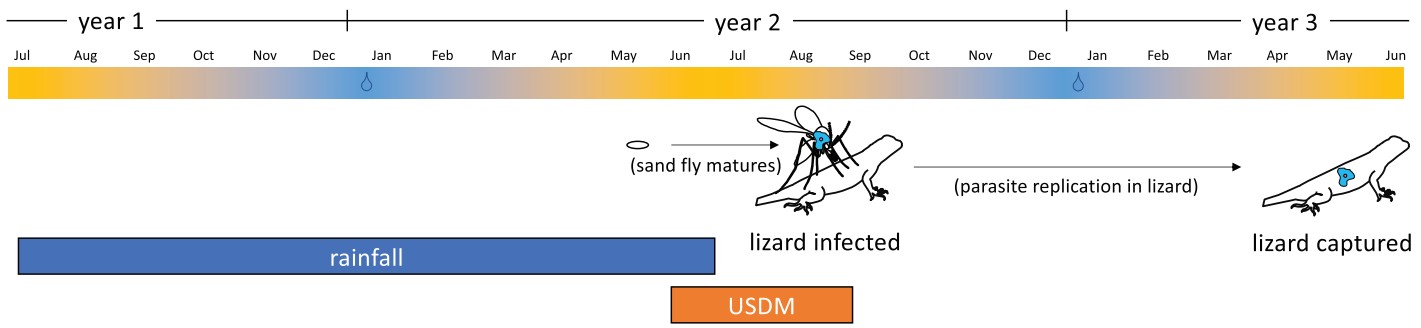

**Figure 1** **Illustration of dates selected for drought data relative to lizard's capture date and presumed date of infection.** The timeline at the top is color coded to show the times of year that are most rainy (blue) *vs* dry (yellow) in this environment. Data on drought severity (USDM) for modeling were summed over the timeframe indicated by the orange box relative to the lizard's capture date. Data on rainfall were summed over the timeframe indicated by the blue box, again relative to the lizard's capture date. Lizards were presumed to have become infected the summer prior to their capture (see text for details). Sand fly generation times are approximately 1.5–2 months, with egg laying peaking in May and late July/August (*Chaniotis & Anderson, 1968*).

drought), D0 (abnormally dry), D1 (moderate drought), D2 (severe drought), D3 (extreme drought) and D4 (exceptional drought). Data are reported online weekly and were converted to a numeric scale for our analysis, with 0 indicating no drought, 1 indicating abnormally dry conditions (D0), and so on, giving a final range of 1 to 5. The data analyzed were from Ukiah, Mendocino County, California, which is the closest USDM location to HREC (about 30 km).

Because we have data on infections that predate the formation of the US Drought Monitor, we also used an additional measure of water availability to enable the analysis of earlier prevalence data: rainfall data collected from HREC Headquarters. These data were collected starting in the 1950s and are still being collected today, providing a long-term, localized picture of the rainfall patterns over multiple decades.

## Selection of data for modeling

To get an accurate picture of how prevalence and the number of clones per infection change over time, we wanted to ensure that data included in the analysis were taken from sites that were sampled regularly over the course of this study. For that reason, we removed infections from any sites that were not sampled at least 10 years since 1978.

Next, we used our knowledge of the parasite's biology to select the drought dates that were most likely to influence a lizard's infection status and COI (Fig. 1). Most lizards included in this study were collected early in the summer (10,411 of 14,011 lizards-almost 75%-were collected in May or June). Given that most infections begin mid-summer (*Bromwich & Schall, 1986*) and that infections of *P. mexicanum* are believed to be chronic, most infections observed in early summer were likely transmitted to the lizard the previous summer. Additionally, lizards are not long-lived, so they are unlikely to be captured two years after their infection date. Considering these observations, we chose to structure our analysis under the assumption that lizards with a patent infection became infected the summer prior to their capture. We expected drought might influence COI and prevalence *via* a change in transmission, so the date ranges included in the models for all measures of

**Table 1 Summary of the variables included in the models for statistical analysis.**

| Data | Model |
|---|---|
| 2010 to 2016 | Infection status ~ drought category + sex + size + site |
| 2010 to 2016 | COI ~ drought category + sex + size + site |
| 2001 to 2016 | Infection status ~ USDM + sex + size + site |
| 2001 to 2016 | COI ~ USDM + sex + size + site |
| 1978 to 2016 | Infection status ~ rainfall + sex + size + site |
| 1996 to 2016 | COI ~ rainfall + sex + size + site |

Note:
'Data' indicates the years from which data were drawn for a given model. 'Model' summarizes the outcome (before '~') and predictor (after '~') variables included in each model, where 'infection status' indicates whether the lizard is infected (1) or not (0), 'COI' is a measure of the number of distinct clones carried by an infected lizard (see text for details), 'drought category' is a categorical variable indicating whether the lizard was collected ('pre') or during/after ('post') the drought, 'sex' indicates the lizard's sex ('m' or 'f'), 'site' indicates the site from which the lizard was collected (see Table 2 for site codes), 'USDM' is a continuous variable indicating the United States Drought Monitor's average drought designation for the summer prior to the lizard's capture (0 = no drought, 1–5 correspond to increasing severity of drought, see text for details), and 'rainfall' is the total rainfall in millimeters for the rainfall year (July to June) ending the year prior to the lizard's capture.

drought were chosen to best reflect water availability during the summer prior to the lizard's capture date (Fig. 1). As stability of COI within infections has been documented on this system, using this time period for infection is appropriate (*Hicks & Schall, 2014*). For the USDM data, we averaged the drought severity for the summer months a year prior to the lizard's capture (*i.e.*, for a lizard captured in 2001, we averaged drought severity June 2000–August 2000). For rainfall data, we used the total rainfall in the rainfall year ending the summer prior to the lizard's capture (*i.e.*, for a lizard captured in 2001, we used the total rainfall July 1999–June 2000). Figure 1 illustrates how the drought measures correspond to the lizard sampling date and presumed date of infection.

## Analysis

We analyzed our data on three different time scales: immediate, mid-length (dates determined by availability of USDM data), and long-term (dates determined by availability of prevalence/genetic data; Table 1). For all models, drought was the main predictor variable and the lizards' sex, size (SVL) and site of capture were included covariates; sex, size and site have all been shown to influence a lizard's probability of becoming infected in past studies (*Schall & Marghoob, 1995*). For each time scale, two models were constructed: one with COI as the outcome variable (estimated as the maximum number of alleles at any of the four microsatellite loci sampled) and one with infection status as the outcome variable. Infection status was used as a host-level measure of prevalence so that lizard sex and size could be incorporated into the model (prevalence cannot be calculated for an individual lizard, but is the sum of the infection status of every lizard in a region of interest).

First, we tested the immediate effects of the worst drought (2013–2014) by comparing COI and infection status pre-drought *vs* post-drought. This analysis did not use any quantitative measure of rainfall/drought to assess drought, but rather assigned data collected in the three years immediately preceding the drought (2010–2012) to a 'pre-drought' category and the data collected in the three years during and immediately

**Table 2 Summary statistics by collection site for all infection-related data analyzed in this study.**

| Site | # Years | Lizards | % Male | Size | # Genotyped | COI | % Infected |
|------|---------|---------|--------|------|-------------|-----|------------|
| CC | 33 | 1,857 | 55.8 | 63.4 | 32 | 1.94 | 18.0 |
| COY | 20 | 1,746 | 52.4 | 63.3 | 5 | 3.20 | 9.9 |
| GH | 17 | 476 | 55.3 | 61.5 | 54 | 1.63 | 18.1 |
| HC | 29 | 1,169 | 53.7 | 64.1 | 28 | 1.93 | 12.7 |
| JOY | 29 | 1,870 | 51.2 | 62.9 | 26 | 1.50 | 9.5 |
| LLH | 15 | 600 | 51.3 | 63.5 | 0 | NA | 12.7 |
| MLH | 25 | 2,517 | 53.2 | 62.8 | 153 | 1.93 | 19.1 |
| NEW | 14 | 422 | 58.3 | 64.6 | 1 | 1.00 | 10.9 |
| PARS | 29 | 1,990 | 56.4 | 61.8 | 89 | 2.25 | 19.4 |
| WT | 29 | 1,364 | 58.9 | 62.8 | 158 | 2.00 | 26.8 |
| All | 34 | 14,011 | 54.4 | 63.0 | 546 | 1.97 | 16.2 |

Note:
'Site' is a code for the site from which lizards were collected. For each site, we report: '# years', the number of years lizards were collected from that site; 'lizards', the total number of lizards collected from that site over all years combined; '% male', the percent of lizards collected at that site that were male; 'size', the mean size (snout-to-vent length) in millimeters of lizards collected from each site; '# genotyped', the number of infections with genotyping data; 'COI', the average number of clones per infection (complexity of infection); and '% infected', the percent of all lizards collected from each site that were infected.

after the drought to a 'post-drought' category (2014–2016). Unfortunately, due to complications outside the researchers' control, data were not collected in 2013. Data on COI were analyzed by fitting a zero-truncated Poisson (ZTP) model using function *vglm*() with family = pospoisson from R package "VGAM"; uninfected lizards were excluded from this analysis. Previous analysis of infections from individual sites in a single year suggest that the ZTP model is appropriate for modeling COI in this system (*Neal, 2021*). Prevalence data were analyzed using a logistic regression due to the binary nature of the dependent variable (lizards are either infected (1) or uninfected (0)).

Second, we tested whether COI or infection status were correlated with the severity of drought as reported in the USDM for data collected 2001 to 2016. As explained above, our measure of drought was the average drought severity the summer before each lizard was captured. Again, COI data were analyzed by fitting a ZTP model and infection status data were analyzed by logistic regression.

Finally, we tested whether infection status or the number of clones per infection was correlated with rainfall on a longer time scale (the USDM only goes back to 2000). As explained above, rainfall data were summed over the rainfall year (July to June) prior to the lizard's capture year.

Due to some conflicting results between analyses, we repeated the second analyses (those using USDM data) using the more limited range of years used in the first set of analyses (2010–2016) and we repeated the third set of analyses (those using rainfall) using the more limited range of years used in the first and second sets of analyses (2010–2016 and 2001–2016) to help clarify whether apparent conflicts in the results were related to the range of years used or the measure of drought analyzed.

## RESULTS

Of the 26,243 lizards collected from roughly 150 sites in the years 1978 to 2016, 14,011 lizards from 10 sites were included in the final analysis. The remaining lizards were excluded for a number of reasons including having been collected at a site that was not sampled regularly (most common reason for exclusion) and having missing data for one of the variables analyzed (site, SVL, sex and infection status). Table 2 shows a summary of the lizards included in the analysis organized by the site from which they were collected. Because the majority of the samples utilized in the project were included in previous studies, the specific genetic diversity indices for the samples can be found in those references (*Schall & Vardo, 2007*; *Vardo-Zalik & Schall, 2008*; *Vardo-Zalik, Ford & Schall, 2009*; *Fricke, Vardo-Zalik & Schall, 2010*; *Hicks & Schall, 2014*).

The USDM was established in 1999, so USDM drought index values are not available prior to 2000; thus rainfall was used as an alternate measure of water availability for infection data predating 2000. Rainfall data collected at HREC is significantly correlated with the USDM drought index over the years for which both are available (2001–2016; Pearson r = −0.7, *p* = 0.0018, Fig. 2A).

### Analysis 1: before *vs* after the 2013/2014 drought

A total of 2,152 lizards from eight sites (LLH and NEW have no data from 2010 to 2016) were included in our first analysis: 1,176 lizards from before the drought and 976 lizards from during and after the drought. Results from the logistic regression indicate a significant effect of drought category ('pre' *vs* 'post') on infection status while controlling for variation in lizards' collection site, SVL and sex (estimate (pre) = −0.37, z = −2.296, *p* = 0.02, GLM, family = binomial). Holding all else constant, the model predicts a 31% ($e^{-0.37}$ = 0.692) decrease in prevalence before the drought relative to during/after the drought (or, flipped, a 45% ($e^{0.37}$ = 1.45) increase in prevalence after the drought). Collection site, SVL and sex also significantly influenced prevalence (*p* < 0.05 in R's 'drop1. vglm' analysis), which is not surprising given previous findings and is the reason these variables were included as covariates in the model.

Of the lizards mentioned above, genetic data were available for 184 of the 189 infected lizards: 96 pre-drought and 88 post-drought. Analysis revealed no effect of drought category on the COI (estimate (pre) = 0.28, z = −1.356, *p* = 0.175, VGLM, family = pospoisson), and neither did the lizard's collection site, sex or SVL.

### Analysis 2: correlation with drought using USDM data (2001–2016)

A total of 5,842 lizards from 10 sites were included in the second analysis with 124–720 lizards sampled per year (median 382, mean 389.5). Results from the logistic regression indicate no significant effect of the USDM's drought index on lizard infection status when controlling for variation in lizard's collection site, SVL and sex (estimate = 0.048, z = 1.383, *p* = 0.17, GLM, family = binomial). Collection site, SVL and sex again showed a significant effect on prevalence (*p* < 0.0001 in R's 'drop1.vglm' analysis).

Of the 608 infected lizards included in the analysis above, genetic data were available for 460 infections. Most of the missing data comes from earlier years (*e.g.*, only 6/95 infected

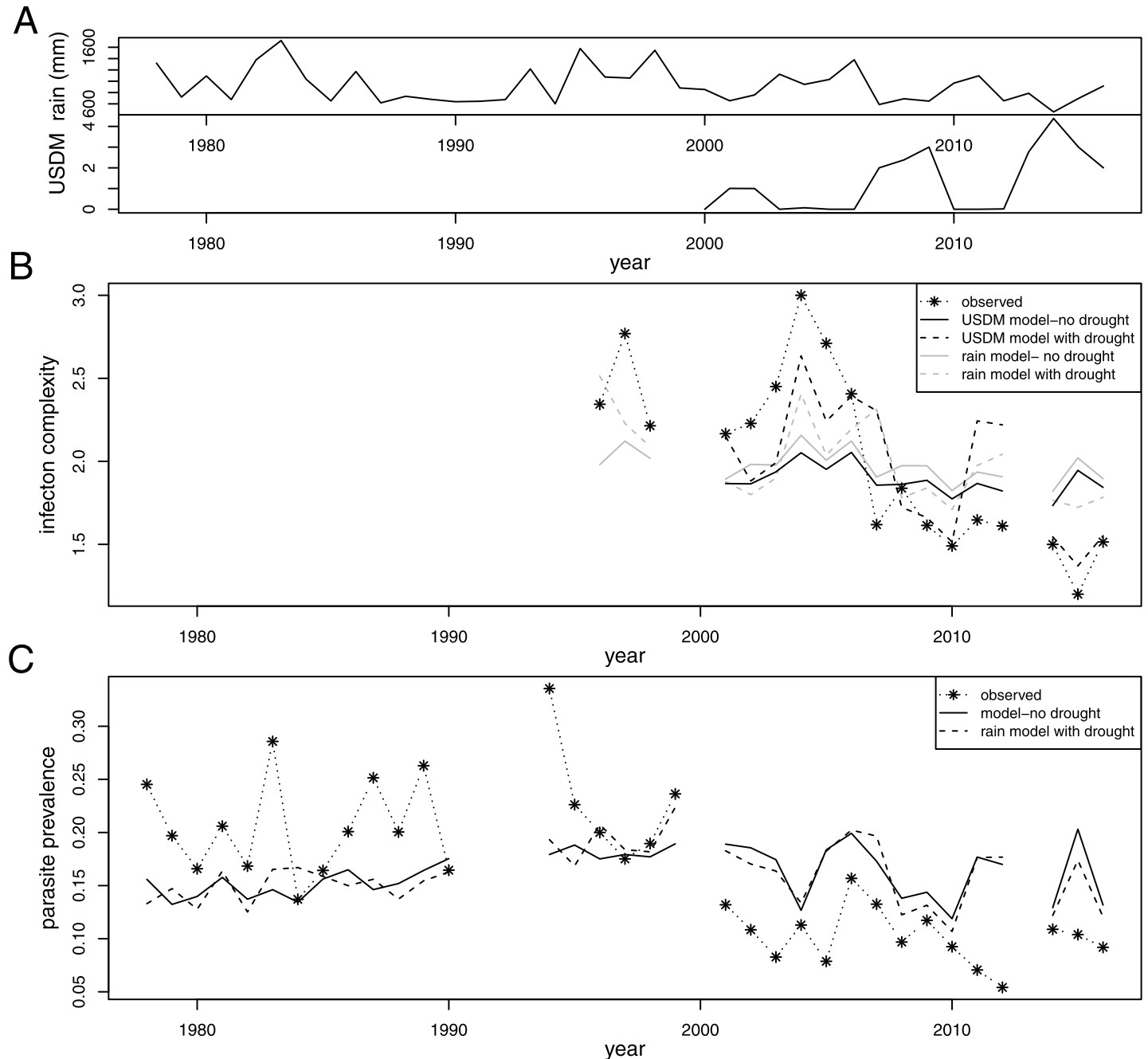

**Figure 2 Summary of drought and infection data in relation to model predictions with and without drought as a predictor variable.** (A) A summary of the drought measures over time: the annual rainfall for the rainfall year (July–June; 'rain') and the average US drought monitor drought rating (available starting in 2000) for the summer months (expressed as a number ranting from no drought = 0 to extreme drought = 5; 'USDM'). (B and C) Observed and modeled infection complexity (COI; data collection started in 1996) and parasite prevalence respectively. The models are shown with and without drought/rainfall as a predictor variable to allow visual comparison of the effects of drought on the model.

lizards from 2001 have genotyping data available) when genetic data were not collected as consistently. Analysis reveals a significant effect of drought on COI (estimate = −0.25, $z = -6.282$, $p < 0.0001$, VGLM, family = pospoisson). The model predicts that the number of clones in an infection will be decreased by a factor of 0.8 ($e^{-0.25} = 0.778$) for every unit increase in the drought index. Figure 2B shows observed numbers of clones per infection over time in comparison with modeled clones per infection with and without drought as a predictor variable. SVL and sex did not affect the number of clones per infection, but the number of clones per infection did appear to be influenced by site ($p = 0.0032$ in R's 'drop1. vglm' analysis).

### Analysis 3: correlation with rainfall using full data set (1978–2016)

When analyzing the full dataset, rainfall showed a significant correlation with infection prevalence (estimate = $3.8e^{-04}$, $z = 5.42$, $p < 0.0001$; GLM, family = binomial). Based on this model, for every 1 mm of additional rainfall, prevalence is expected to increase by about 0.04% ($e^{0.00038} = 1.000386$). For reference with the first model, the average rainfall 2010–2012 was about 250 mm greater than 2014–2016; thus prevalence is predicted to be 10% higher prior to the drought. Overall, rainfall ranged from 392 to 1,718 mm per year, which corresponds to a predicted 50% expected change in prevalence between the lowest and highest rainfall years. Figure 2C shows observed prevalence over time in comparison with modeled prevalence with and without rainfall as a predictor variable. As in previous analyses, the lizards' collection site, SVL and sex also showed a significant correlation with infection prevalence ($p < 0.0001$).

As mentioned in the methods, genetic data was first collected starting in 1996. From 1996 to 2016, a total of 546 lizards in our data set had genotyping data available. While earlier years in this sequence have a higher frequency of missing genotyping data, the greater number of infected lizards collected some of these years means that a substantial amount of genotyping data is still available (*e.g.*, roughly 30 lizards per year in 1996–1998). Data from later years is comparable, with a mean of 30.1 genotyped infections per year from 2010–2016.

Rainfall showed a significant correlation with COI over the full data set (estimate = 0.00062, $z = 5.451$, $p < 0.0001$; VGLM, family = pospoisson). The number of clones per infection is expected to increase by a factor of 1.00062 for every 1 mm increase in rainfall. Figure 2B shows observed numbers of clones per infection over time in comparison with modeled clones per infection with and without rainfall as a predictor variable. As in the previous analysis, site was also a significant predictor of the number of clones per infection ($p = 0.0024$ in 'drop1.vglm' analysis), but lizard sex and SVL were not ($p > 0.05$).

### Comparison of data sets *vs* drought measures

Table 3 shows a summary of the impact of drought on prevalence (infection status) and COI for each combination of drought measure and year range. The relationship between drought and prevalence indicated by our models differs depending on the range of years used for analysis but not depending on the measure of drought used. The relationship

**Table 3 Relationship between drought and parasite prevalence or complexity of infection (COI) as indicated by models incorporating different time scales and drought measures.**

| Model | Data | Drought category | USDM | Rainfall |
|---|---|---|---|---|
| Prevalence | 2010–2016 | + | + | + |
| | 2001–2016 | NA | NS | NS |
| | 1978–2016 | NA | NA | – |
| COI | 2010–2016 | NS | (–) | – |
| | 2001–2016 | NA | – | – |
| | 1978–2016 | NA | NA | – |

Note:
'Model' indicates which outcome variable was modeled for the corresponding results. 'Data' indicates the range of years from which data were taken. The measures of drought used in the model are 'drought category' (pre *vs* post), USDM (the US Drought Monitor's drought index), and 'rainfall'. Within the table, '+' indicates that prevalence or the number of clones (COI, complexity of infection) increased with increasing drought (decreasing rainfall), '–' indicates a decrease in these values with increasing drought (decreasing rainfall), and parentheses indicate a marginally significant relationship ($0.05 \leq p \leq 0.1$). 'NA' indicates that data on the specified drought measure were not available for the full range of years so the analysis was not performed. 'NS' indicates that the specified drought measure did not have a significant impact on prevalence/COI.

between drought and COI indicated by our models does not differ depending on the range of years used for analysis or the measure of drought used with the exception that some year ranges may have too little data to reveal a detectable pattern. Detailed results for models used only for comparison are not shown.

## DISCUSSION

The availability of water in an environment can have a major impact on the ecological processes of many species, including parasites. For this sand fly transmitted malaria parasite, our results suggest that drought is connected to a decrease in COI. Analyses on multiple timescales show a negative relationship between drought/rainfall and the number of parasite clones per infection, and the improvement in the predictive power of the model when drought is included can be observed in Fig. 2B. The pattern, however, is less clear for prevalence. When the full range of years are included in the analysis, increasing drought appears to be related to a decrease in prevalence of *P. mexicanum*, but on a shorter time scale, prevalence appears to be uncorrelated with drought (2001–2016) or even to have a positive relationship with drought (2010–2016; Table 3).

These data provide a rare glimpse into how drought impacts the long-term dynamics of a host-parasite system, but they also present a few puzzles. First, why does the relationship between drought and prevalence depend on the timescale over which the data are analyzed? Second, what might cause COI to be more clearly affected by drought than prevalence? We consider each of these questions below.

### Drought and prevalence

While analyses linking drought and COI showed a consistent negative association over multiple time scales, the data on prevalence are much harder to interpret. When examining only data from the few years immediately surrounding the most recent drought (2010–2016), it appears that *P. mexicanum* becomes more common following this severe dry period. However, when a longer data series is examined-one that includes additional

notable drought events (*Swain et al., 2014*)-this pattern is either not apparent (2001–2016) or reversed (1978–2016; Table 3).

While the ambiguous results concerning drought and prevalence could be due to our methodological approach, we do not feel that this is a likely explanation. For example, inconsistencies in the direction of a correlational effect are not unheard of, and may occur if the 2013/2014 drought was so exceptional as to have an effect on the prevalence of *P. mexicanum* unlike the typical effects of a low rainfall year. It is well recognized that the effects of certain biotic and abiotic factors may be non-linear (*e.g.*, *Otero et al., 2019*). We chose not to include a non-linear rainfall term in our models because we were interested in how drought specifically (not rainfall in general) impacted parasite prevalence; by definition, drought occurs on the extreme low end of the range in rainfall, so we did not feel that non-linear relationships between rainfall and prevalence were relevant to our current study. That said, it is possible that moderate drought and extreme drought affect prevalence differently, which could explain the apparent contrasting effects of drought in our short *vs* long term data sets.

It is also possible that certain effects of drought may occur after a time lag, which could have been captured in the 'categorical' analysis but not in the 'continuous' analysis. For example, drought may decrease or have no immediate effect on vector populations but could then cause a surge in their numbers the following year (reviewed in *Brown, Medlock & Murray (2014)*). When performing the analysis using drought/rainfall as a continuous variable, we did our best to carefully select a period of rainfall/drought data that would most likely impact prevalence (see Methods). While its possible that this may not have captured the relevant period as well as a simple "before *vs* after" categorical analysis, a visual examination of the data suggests this is not the answer; the prevalence of *P. mexicanum* is consistently higher all three years after the 2013/2014 drought than it was prior to the drought.

Perhaps the most likely explanation for the contrasting results between the short and long timescale is that something other than the drought caused a change in prevalence in 2010–2016. Examining the data on annual prevalence in Fig. 2C, it appears that prevalence was uncharacteristically low prior to the 2013/2014 drought. This may have been a temporary dip in prevalence that is unexplained by any of our measured variables, and thus what appears in our analysis as an increase in prevalence after this drought may instead be a return of prevalence to its "normal" range. This is a common and widely recognized pitfall of observational data, so much so that it has been acknowledged by the commonly heard phrase "correlation does not imply causation". Any number of biotic and abiotic factors could have caused this extremely low prevalence in 2011–2012. Some malaria species, including *P. mexicanum*, appear to show unexplained cycles in their prevalence (some examples reviewed in *Schall (2000)*), and this drought may simply have aligned with a change in cases that would have occurred regardless of the drought. Even though this is a well-known limitation of observational data, especially data taken from a single site surrounding a single event, our results highlight the importance of treating such data with caution. We believe this underscores the significance of long-term studies even more; long-term data sets have a greater ability to place the apparent effects of unusual

events into context, making it easier to assess whether any observed changes were actually caused by a specific focal event.

On the longest time scale (1978–2016), there appears to be a negative association between drought and prevalence. This period included two notable droughts (1987–1992 and 2013–2014) and immediately followed another drought in 1976–1977 whose impact, if any, should have been noticed in 1978 (drought dates from *Swain et al. (2014)*). Despite the statistical significance of rainfall in the models of prevalence spanning the full range of years, inclusion of this term does little to improve the model fit (Fig. 2C). If there is a true causal relationship between drought and prevalence, it is not strong, and the effect may often be overcome by other factors that show a larger impact on the system.

## Drought and infection complexity

We originally chose to analyze the effects of drought on infection complexity because we expected that drought would decrease transmission, reducing both prevalence and COI. To better understand this expected connection, let us review a simple model of transmission as it relates to the distribution of malaria clones among infections. If all hosts at a site have equal probability of being bitten by a vector, we would expect infectious bites to be distributed among hosts following a Poisson distribution. This distribution pattern is used in many models of malaria multiplicity of infection (MOI; *e.g.*, *Hill & Babiker, 1995*; *Schneider & Escalante, 2014*; note that MOI is similar to COI except that it accounts for every clone that enters a host-even superinfection with an identical clone-while COI counts only unique clones; this distinction is summarized in *Neal, 2021*). If the number of infectious bites increases, this should increase both the total number of hosts bitten (prevalence) and the number of hosts bitten multiple times (MOI, and COI providing many different clones are circulating, which appears to be there case in this system). Following this model, prevalence and infection complexity should both be affected if some factor (*e.g.*, drought) alters transmission, such as by reducing the population size of vectors.

Interestingly, drought appeared to have a more consistent effect on COI than on prevalence. Infection complexity, therefore, may be reduced during drought periods by something beyond a simple reduction in transmission events, such as competition within the lizard or sand fly host for resources. Prior research on human, rodent and lizard malaria all suggest that multiclonal infections may be established collectively, with a single infectious bite transferring multiple clones (*Druilhe et al., 1998*; *de Roode et al., 2005*; *Vardo, Kaufhold & Schall, 2007*; *Vardo-Zalik, 2009*). Drought could reduce the establishment of multi-clone infections in the vertebrate host without affecting prevalence if drought placed non-lethal stress on vectors that made them less capable of supporting the development of multiple malaria clones. If this were the case, water-stressed sand flies might be less capable of collective transmission than sand flies that experienced more accommodating humidity levels. Similarly, water-stressed lizards might present a more hostile or resource-limited environment to the parasites they host, increasing competition between co-infecting clones and reducing the maintenance of multi-clone infections. We do not know of any empirical evidence supporting this possibility, but it may be an interesting avenue for future research.

## CONCLUSIONS

This study provides valuable data on how drought impacts both prevalence and COI from a well-studied natural system over a long period of time. This data not only bolsters existing data on the impacts of drought on vector-borne disease, but additionally suggests a few general points that should be considered in future research.

First, COI should not be ignored in studies of drought and other environmental influences. We found a significant negative association between drought and the number of clones per infection over multiple time scales. While our study is not able to identify the cause of this association, one possibility is that water stress affects the ability of multi-clone infections to be maintained in either the vertebrate host or vector. This hypothesis could be explored in future research, and we encourage other researchers to investigate the effects of drought on COI for additional malaria species or other vector-borne parasites. Infections with fewer genetic lineages have been linked with higher virulence and a reduced ability to resist superinfection in *Plasmodium falciparum* (reviewed by *Smith et al. (1999)*), suggesting that drought may cause shifts in human disease outcomes independent of any changes to prevalence.

Second, we must continue to be cautious of data taken from a limited time period. Our data, at least in reference to parasite prevalence, showed very different results when analyzed on a short *vs* long time scale. This cautionary tale is nothing new or surprising but should always be remembered when considering observational data of limited scope.

Finally, it should not be assumed that prevalence and complexity of infection will be impacted in the same way by environmental perturbances. If it is assumed that environmental stressors will primarily impact transmission and that infections with multiple clonal lineages are always a result of multiple infectious bites, it might seem unnecessary to examine impacts on both clonal diversity and prevalence. However, our results suggest that these two outcomes can be affected differently, and neither should be ignored. The semi-independence of these outcomes may also suggest that drought (and perhaps other abiotic factors) may impact parasite systems in ways that are more complex than a 'simple' change in transmission intensity.

## ACKNOWLEDGEMENTS

We are grateful to the staff at the University of California's Hopland Research and Extension Center for their continuing hospitality in supporting this research over multiple decades. We also acknowledge the countless hours of effort by Jos. J. Schall and members of the Schall, Vardo-Zalik and Neal labs in collecting and recording this long-term data set.

### Funding

This work was supported through internal funds from the Pennsylvania State University, York campus for Anne Vardo-Zalik. Joshua Sassi was supported by the Countryman Fund, Chase Endowment (Chase Student Research), and a Weintz Fellowship from Norwich University. Allison T. Neal was supported by the Chase Endowment (Chase Faculty) from

Norwich University. The funders had no role in study design, data collection and analysis, decision to publish, or preparation of the manuscript.

### Grant Disclosures

The following grant information was disclosed by the authors:
Pennsylvania State University, York.
Countryman Fund, Chase Endowment (Chase Student Research), and a Weintz Fellowship from Norwich University.
Chase Endowment (Chase Faculty) from Norwich University.

### Competing Interests

The authors declare that they have no competing interests.

### Author Contributions

- Allison Neal conceived and designed the experiments, performed the experiments, analyzed the data, prepared figures and/or tables, authored or reviewed drafts of the article, and approved the final draft.
- Joshua Sassi conceived and designed the experiments, performed the experiments, authored or reviewed drafts of the article, and approved the final draft.
- Anne Vardo-Zalik conceived and designed the experiments, performed the experiments, analyzed the data, prepared figures and/or tables, authored or reviewed drafts of the article, and approved the final draft.

### Animal Ethics

The following information was supplied relating to ethical approvals (*i.e.*, approving body and any reference numbers):
    IACUC, Penn State University.
    IACUC, University of Vermont.

### Field Study Permissions

The following information was supplied relating to field study approvals (*i.e.*, approving body and any reference numbers):
    Field studies were approved by the Hopland Research and Extension Center, Hopland, CA. Scientific Collecting Permits issued by the state of CA were utilized throughout the course of this study, with the most recent years covered under permit SC-7971.

### Data Availability

    The data and R-code are available at OSF and Zenodo: Neal, Allison T. 2023. "Drought, COI and Prevalence of *Plasmodium mexicanum*." OSF. January 2. osf.io/ztj8s.
    Allison Neal, Anne Vardo-Zalik, & Joshua Sassi. (2023). Drought, infection complexity (COI), and prevalence of *Plasmodium mexicanum* [Data set]. Zenodo. https://doi.org/10.5281/zenodo.7499450.

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
