# Peer review of "Drought correlates with reduced infection complexity and possibly prevalence in a decades-long study of the lizard malaria parasite Plasmodium mexicanum"

_PeerJ, doi:10.7717/peerj.14908_

## Round 0.1 · original submission · Major Revisions

The authors studied a large dataset and this is commendable. Two reviewers are of the opinion that the data analysis is suboptimal and the authors should pay attention to these comments as well as the comment about the labelling of the data should be dealt with as this would impact on repeatability in future studies. The lack of recent literature, particularly referring to global warming should also be addressed.

In conclusion, all three reviewers provided comprehensive and fair comments. Provided that you address these the manuscript may be resubmitted after addressing a major revision.

Reviewer 1 ·

Basic reporting

No comment

Experimental design

No comment

Validity of the findings

Neal and colleagues present a study on genetic complexity of lizard malaria over time. While the data set is interesting, several key points need clarification. My comments focus on main points that need to be addressed:

1) The manuscript is lacking a lot of crucial detail that should be standard in epidemiological studies:
Number of lizards sampled, number infected, number typed (also needs to be in the abstract. See e.g. https://peerj.com/articles/13485/ for a good example)
Diversity (number of alleles, expected heterozygosity) of microsatellites. What were the inclusion criteria for minority clones?
I also recommend adding point estimates for each analysis. The model outputs can be difficult to interpret.

2) The manuscript lacks appropriate referencing. Few references published in the last 5-10 years are included, while many referenced studies are old. Given the massive leaps of both genetic diversity studies and research on climate change, more recent references are needed to out the research in context. As example, on lines 316-317, the statement “Extreme weather patterns like drought are likely to become increasingly common with climate change” is referenced with two studies from 2000 and 2006. More recent research needs to be published.

3) Lines 26-28: The authors state “To our knowledge, this is the first reported evidence for environmental factors such as drought affecting the abundance of multi-clonal infections in malaria parasites.” I do not think this is correct. Several studies have compared COI of human infections between the wet and the dry season.

4) Please add some detail on what the drought levels 1-4 mean. Could it be that at any levels of drought mosquito populations are greatly diminished, thus explaining the lack of associations?

5) The authors state “Given that most infections begin mid-summer (Bromwich and Schall, 1986) and that infections of P. mexicanum are believed to be chronic, most infections observed in early summer were likely transmitted to the lizard the previous summer” and analyze the data accordingly. This is a major weakness that needs discussion and further analysis. They cite a single study stating that infections persist that long. In human infections, it is observed that COI decreases over time. At the very least, the authors should run the model using the drought status 1-3 months prior to sampling as predictor.

6) The discussion is too long and often highly speculative, e.g.:
Lines 347-410: I would claim that these sections are not even driven by a hypothesis but pure speculation. I recommend shortening them to 2-3 sentences.
Lines 431-460: These sections are very hard to follow and I am not sure what the authors want to say. I believe a simple sentence can do then job: “Further research is required on why droughts seem not to impact prevalence”.
The discussion leaves the impression, that even the authors are not sure what they found (“Our own feeling is that...”, “Multiple possible explanations exist…” etc.)
The conclusions are way too long. Conclusions should be a succinct paragraph, focusing on the findings of the study presented.



Minor comments:

Lines 356-357: ”note that MOI is similar to COI except that it accounts for every clone that enters a host- even superinfection with an identical clone- while COI counts only unique clones”
Please provide a reference for this statement, I never heard this definition.

Lines 339-341: “drought appears to be uncorrelated with drought (2000 – 2016) or even to have a positive relationship with drought (2010 – 2016; Table 3).” ???

Additional comments

No comment

·

Basic reporting

This manuscript describes the effect of drought to infection complexity and the prevalence of lizard malaria parasite of Plasmodium mexicanum by utilizing long span data set. Authors suggest that drought could decrease complexity of infection but give no clear impact to parasite prevalence. No enough explanation for those observed results has been proposed yet, however, this study provides a quite impact that environmental factor can affect malaria parasite infection complexity for the first time with association to recent climate change.

Experimental design

no comment

Validity of the findings

Obtained results are very significant to understand vector borne infectious diseases deeply and more suggestive for future study in this area.

Additional comments

Therefore, the reviewer recommends that this manuscript can be accepted for publication in this journal after minor revision according to following comments.

Line 73. CA should be fully described.

Tables.
All tables can be edited according to a standard format by using necessary lines between data sets.

Table 2.
Those item names of each subject can be reconsidered for easy understanding and this table should be adequately edited.

Figure 2.
Detailed legends of Y axis of Figure 2A and 2B should be modified. ‘usdm’ appeared in Fig. 2A should be ‘USDM’ according to the figure legend. Also ‘min. clones’ in Fig. 2B should be kindly modified.

Reviewer 3 ·

Basic reporting

Neal and colleagues analyzed an impressive number of lizard blood samples to explore how malaria parasites prevalence and clone diversity correlates with seasonal droughts over the year. This enabled authors to draw conclusions that would have been missed if only narrow cohorts were assessed instead. The study is well written and is pleasant to read. I don’t have major concerns about the manuscript, but I believe that some points regarding reporting of results and interpretation should be addressed before publication.

I am not sure if the raw data provided would make it possible to other researchers to reanalyze the data or to use it in other studies. For example, the "DroughtAnalysisDataFull" file do not have the meteorological data. Some dates seem to be wrong (e.g., there are a few "20010500" there) and it would be tricky to link the results from each sample to the information regarding rainfall and drought that were provided in different files. It may be worth finding a way to have all relevant information in a single .csv, or authors need to make sure information from all three .csv files are correct.

Experimental design

The data analysis and modeling seem to be robust to test the effects of drought on parasite prevalence and clonal diversity. Analyzing such long-spanned data created some difficulties such as missing sampling time points and meteorological data. This latter point would be a concern, but I think authors rationale to use rainfall instead of drought is methodologically sound because these factors are correlated during the time when both type of information was collected.

Validity of the findings

Regarding the conclusions drawn, I am not sure if authors can say that prevalence has a “positive relationship with drought” when the 2010-2016 data was analyzed (lines 340-341). This is because authors use the term “drought” in relation to the “regular” California drought cycle, which was tested using the 2000-2016 data. For the 2010-2016 cohort, models tested the effects of a specific drought only (the 2013-2014 one) and found increased prevalence during/after the drough. Therefore, using “drought” (line 341) referring to the specific 2013-2014 one, right after using the same term in line 340 that relates to the “general drought cycle”, may be misleading. To me it seems that the general long-term pattern is that prevalence is negatively correlated with drought, and that prevalence was higher only after a specific severe drought.

However, Fig. 2b shows that prevalence in years 2011 and 2012 seem to be lower than the historical rates, while rates for the 2014-2016 period seem to be within the historical range. Would it make sense to test whether prevalence was lower before the drought in relation to the the period before and after 2011-2012? This could reveal, for example, that prevalence did not increase after the 2013-2014 drought, but instead, that prevalence was “abnormally” low in the years that preceded the 20132-14 drought.

According to the modeling methods (lines 198-208), authors did not model parasite prevalence and COI in response to USDM scores for the data immediately before and after the 2013-2014 drought (first model). If that is the case, why does table 3 include results for prevalence and COI for 2010-2016 dataset in the USMD column? Please correct or explain how authors found these results.

The USDM scores range from 0 to 4 (as explained in lines 150-152), but for Fig. 2 legend authors used a 0 to 5 range. Because the graph plotted also seems to include 5 (the USDM curve touches the main line, which is above the line related to USDM = 4), it would be important to make sure the ‘usdm’ variable range in the second model was 0-4, and not 0-5.

Additional comments

Instead of simply stating that complex infections can have implications for disease progression and pathogen virulence (lines 44-6), authors could be more specific by showing which implications those might be. This would improve the connection between this paragraph and the one that follows.

Line 98 - Although SVL is a standard measurement in herpetology, maybe authors could briefly explain why this information is collected since PeerJ is a journal that target broad audiences.

It feels like the first paragraph of the Discussion is basically repeating information presented in the Introduction. I think that all of it could be summarized in 1-2 sentences emphasizing the information presented in lines 328-331.

I like figure 1, it helps to understand the rationale explained in the text (lines 166-182). But I think the first segment of the timeline should be “year 1”, and the second and third segments should be “year 2” and year 3”, respectively.

Table 1: I think that, for the first set of models, authors analyzed data collected from 2010 to 2016, not 2012-2016 as it’s written. And authors probably meant to have years 2001-2016 for the second model. Please, double check this.

Line 259 – Please add the information that this section analyzed the 2001-2016 data.

Lines 339-340 – I think the drought in “drought appears to be uncorrelated with” should be “prevalence”.

---

## Round 0.2 · accepted · Accept

Two reviewers are satisfied that their concerns have been addressed. I request that the authors include their comment about the lack of IACUC permission numbers in the methods section of the manuscript prior to the proofs being prepared. It is in the interest of the journal and the authors to make it clear that the collection has been done in accordance with internationally accepted guidelines.

·

Basic reporting

This revised manuscript adequately reflects all my comments and authors also properly rebut. Now this article can be published in this journal after checking the format.

Experimental design

no comment

Validity of the findings

no comment

Additional comments

no comment

Reviewer 3 ·

Basic reporting

This manuscript is a significant contribution to science and is now suitable for publication in its current form. I would like to thank the authors for making the whole dataset available via OSF.
Kind regards,
Francisco C. Ferreira

Experimental design

no comment

Validity of the findings

no comment